# Sensitivity of discrete symmetry metrics: Implications for metric choice

**Allen Hill**, **Julie Nantel** *

School of Human Kinetics, University of Ottawa, Ottawa, Ontario, Canada

* jnantel@uottawa.ca

## Abstract

Gait asymmetry is present in several pathological populations, including those with Parkinson's disease, Huntington's disease, and stroke survivors. Previous studies suggest that commonly used discrete symmetry metrics, which compare single bilateral variables, may not be equally sensitive to underlying effects of asymmetry, and the use of a metric with low sensitivity could result in unnecessarily low statistical power. The purpose of this study was to provide a comprehensive assessment of the sensitivity of commonly used discrete symmetry metrics to better inform design of future studies. Monte Carlo simulations were used to estimate the statistical power of each symmetry metric at a range of asymmetry magnitudes, group/condition variabilities, and sample sizes. Power was estimated by repeated comparison of simulated symmetric and asymmetric data with a paired t-test, where the proportion of significant results is equivalent to the power. Simulation results confirmed that not all common discrete symmetry metrics are equally sensitive to reference effects of asymmetry. Multiple symmetry metrics exhibit equivalent sensitivities, but the most sensitive discrete symmetry metric in all cases is a bilateral difference (e.g. left—right). A ratio (e.g. left/right) has poor sensitivity when group/condition variability is not small, but a log-transformation produces increased sensitivity. Additionally, two metrics which included an absolute value in their definitions showed increased sensitivity when the absolute value was removed. Future studies should consider metric sensitivity when designing analyses to reduce the possibility of underpowered research.

## Introduction

Assumptions of bilateral symmetry in the lower limbs during healthy gait are implicit to much of gait research, and many studies collect and report data from only one side of the body, or report the average both sides [1]. Nevertheless, Sadeghi et al. [1] found evidence for the presence of both symmetric and asymmetric characteristics of kinematic and kinetic aspects of gait among healthy people. Furthermore, several populations, including those diagnosed with Parkinson's disease (PD) or who have experienced stroke, have unilateral neural pathologies which cause distinct gait asymmetries compared to healthy controls [2–6]. These gait asymmetries are of interest in pathological populations for tracking disease progression or rehabilitation progress [6–8].

**Data Availability Statement:** The software and data produced and analyzed during this work are openly available from the Zenodo data repository at https://doi.org/10.5281/zenodo.5396401.

**Funding:** This work was supported by the Natural Sciences and Engineering Research Council of

Canada (https://www.nserc-crsng.gc.ca) [RGPIN-2016-04928 to J.N., RGPAS 493045-2016 to J.N.], and by the Ontario Ministry of Research, Innovation and Science (https://www.ontario.ca/page/early-researcher-awards) Early Researcher Award [ER 16-12-206 to J.N.]. The funders had no role in study design, data collection and analysis, decision to publish, or preparation of the manuscript.

**Competing interests:** The authors have declared that no competing interests exist.

Measurements of gait asymmetry can be grouped into two classes, discrete and continuous metrics, which either compare two scalar numbers or two continuous time-series. In this paper, we will focus primarily on the more common class of symmetry measures: discrete symmetry metrics (DSM). Sadeghi et al. [1] and Viteckova et al. [9] published reviews of literature analyzing gait symmetry which identified five discrete symmetry metrics commonly used in the literature [10–14].

The first of these five discrete symmetry metrics, a ratio, was introduced by Seliktar and Mizrahi [10] for analyzing asymmetry of ground reaction forces. Robinson et al. [11] also measured symmetry in ground reaction forces with a new metric, commonly referred to as the symmetry index. Vagenas and Hoshizaki [12] introduced a third discrete symmetry metric while analyzing the asymmetry of lower limb kinematics. Plotnik et al. [13] introduced a new metric to assess the relationship between gait asymmetry and freezing of gait in PD. Another metric was proposed by Zifchock et al. [14] to address issues of previous DSMs related to the choice of reference value [11]. Additionally, a research group from Newcastle University uses another metric on a number of spatiotemporal gait variables [15, 16], and two new DSM have been recently proposed by Queen et al. [17] and Alves et al. [18]. Despite their conceptual equivalence, the different defining equations for all of these DSM produce unique numeric results which are not directly comparable [19].

Likewise, these metrics produce different standardized effect sizes and findings of significance for the same underlying effect of asymmetry [17–20]; these results suggest that metrics are not equally powered to detect effects of asymmetry [21]. Achieving adequate power (e.g. power > 0.8) is an important factor in experimental design [21], therefore, understanding the differences in DSM sensitivity to effects of asymmetry may allow the design of better powered experiments. No previous studies comparing common DSM were designed to directly assess metric sensitivity, in terms of power to detect an effect of asymmetry.

Statistical power is a function of sample size, significance criterion, and the effect size [21]. A power analysis known as "sensitivity power analysis" in G*Power [22] can analytically calculate power for many statistical tests given a significance criterion, sample size, and effect size. However, symmetry metric sensitivity cannot be assessed in this way because it remains unclear how a given underlying effect of asymmetry translates to standardized effect sizes for each symmetry metric [18]. Additionally, the sensitivity of various symmetry metrics cannot be precisely assessed using (finite) experimental data as a reference effect of asymmetry because observed effect sizes have inherent error, as point estimates, which would propagate to estimates of sensitivity [21]. Data from observational studies which lack known true effects, as in some previous studies [17, 19, 20], are particularly ill-suited for comparing symmetry metric sensitivity because false positives could overestimate metric sensitivity.

To ensure applicability to a range of populations and study designs, the sensitivity of symmetry metrics should be assessed on a wide range of effects. Precise estimates of power require large amounts of data that could be prohibitive to collect experimentally. A Monte Carlo simulation is a convenient method to generate the large volume of data needed to accurately estimate power while ensuring that a broad range of factors of effect size (e.g. mean difference and group/condition variability) are evaluated.

Therefore, as previous studies have covered the literature with respect to the validity of general assumptions about the degree of asymmetry present in gait [1], and with respect to the breadth of commonly used symmetry metrics and data analyzed for gait symmetry [9], the purpose of this study is to provide a comprehensive assessment of the sensitivity of common discrete symmetry metrics using power simulations and unreported data from a previous study [23] to validate simulation assumptions. This study will aid the design of future studies

by informing authors which metric(s) are appropriate to maximize the power to detect asymmetry for their experimental design.

## Materials and methods

### Power simulation

Discrete symmetry metrics are all mathematical functions which accept two arguments (which can be any set of bilateral variables, e.g., left and right step swing time, affected and unaffected joint range of motion, etc.). We represent a generic symmetry function as $S(x, y)$. We assume that the inputs $x$ and $y$ are independent, normally distributed (an assumption made by many parametric tests), have the same sign, and have the same shape with a variability σ (e.g., group, condition):

$$\boldsymbol{\mathcal{X}} \sim x\boldsymbol{\mathcal{N}}(1, \sigma^2)$$
$$\boldsymbol{\mathcal{Y}} \sim y\boldsymbol{\mathcal{N}}(1, \sigma^2)$$

(1)

$$S(\boldsymbol{\mathcal{X}}, \boldsymbol{\mathcal{Y}}) = S\big(x\boldsymbol{\mathcal{N}}(1, \sigma^2),\ y\boldsymbol{\mathcal{N}}(1, \sigma^2)\big)$$

(2)

As noted by Alves et al. [18], a key characteristic of symmetry metrics is the relative difference between $x$ and $y$, the ratio $y/x$. Therefore, Eq (2) can be simplified to

$$S\big(x\boldsymbol{\mathcal{N}}(1, \sigma^2),\ y\boldsymbol{\mathcal{N}}(1, \sigma^2)\big) = S\big(\boldsymbol{\mathcal{N}}(1, \sigma^2), R \cdot \boldsymbol{\mathcal{N}}(1, \sigma^2)\big)$$

(3)

Where $R = y/x$ and represents the asymmetry magnitude. Eq (3) describes the general form that was used for randomly sampling data (parameterized by $R$ and $\sigma$) and subsequent evaluation by a symmetry metric $S$.

Discrete symmetry metrics from eight previously published papers [10–14, 16–18] were included, based on presence in previous reviews, number of citations, or if recently proposed as improvements on previous metrics. Two metrics [13, 16] are defined with absolute values and solely assess asymmetry magnitude; to enable consistent comparisons between metrics, these two metrics were assessed with and without the absolute value applied. As well, Alves et al. [18] proposed a weighting factor for their metric based on the standard deviation of the input data; the weighted and unweighted metric were evaluated. The standard deviation used in the weighting factor was the $\sigma$ value from the set of parameters for a given test, which will produce a best-case estimate of power, as the true standard deviation would not be known for real data.

Simulated data was randomly sampled with standard deviation $\sigma = [0.01, 1]$, and with an asymmetry magnitude of $R = [1, 5]$. This range of parameters is expected to be sufficient to extrapolate metric behavior to untested parameters. Statistical power can be estimated as the proportion of significant test results when comparing simulated symmetric data (where $R \cong 1$) to asymmetric data (where $R \neq 1$) using a paired $t$-Test. Significance was defined as $\alpha = .05$, and test sample sizes of $N = [10, 100]$ were evaluated. For every combination of testing parameters (R, σ, sample size N), all metrics were tested on the same randomly sampled data, to prevent extreme values from unevenly affecting metrics, and power was estimated from the proportion of 20 million tests. The rate at which metrics' power increases versus asymmetry magnitude distinguishes metric sensitivity. Sufficient power was defined as >0.8, and a difference between metrics of >0.1 in R at the 0.8 power threshold was deemed a practically significant difference in metric sensitivity.

## Experimental data

A previously published study from these authors used a split-belt treadmill to mechanically induce asymmetric gait [23]. Kinematic data were collected at a frequency of 100 Hz using the Computer Assisted Rehabilitation Environment (CAREN; CAREN-Extended, Motekforce Link, Amsterdam, NL) which includes an instrumented split-belt treadmill (Bertec Corp., Columbus, OH) and a 12 camera Vicon motion capture system (Vicon 2.6, Oxford, UK). Gait speed during tied-belt walking was set at 1.2 m/s, while during split-belt walking, the left belt maintained the 1.2 m/s speed and the right belt was slowed to 80% of the left belt, 0.96 m/s. Gait trials lasted 200 s, and the first 25 s were ignored to ensure participants had reached a steady-state. Gait events were calculated using an algorithm based on the local extrema of the vertical position and velocity of the heel marker [24]. Swing time was calculated in units of per-cent stride; future references to swing time will omit mention of the units. Swing time asymmetry was compared between tied and split-belt walking, as previous studies show that split-belt walking causes immediate changes in stance and swing time [25]. Swing time asymmetry was evaluated using all metrics. For comparison to modelling assumptions made in the power simulations, normality of left and right swing times were assessed using the Anderson-Darling test, and Bartlett's test was used to test equality of variances between left and right sides. Tied and split-belt conditions were compared using a paired $t$-Test. All statistical tests used $\alpha = 0.05$. All simulations and analyses were performed using the Julia programming language [26] using custom code [27].

## Results

### General metric characteristics

All discrete symmetry metrics evaluated here are shown in Table 1, including their defining equations and important characteristics (the value for perfect symmetry, the limits of the functions for positive inputs, and whether the metric has a directional output—an output which uniquely signifies which input, x or y, is larger/smaller). The order of x and y arguments have been adjusted, when necessary, such that a larger y always produces a positive asymmetry

**Table 1. Discrete symmetry metrics.**

| Metric | Definition | Perfect symmetry | Limits[†] ($x\rightarrow\infty, y\rightarrow\infty$) | Directional |
|---|---|---|---|---|
| Seliktar and Mizrahi (Sel86) [10] | $S(x,y) = \frac{y}{x}$ | 1 | $0,\infty$ | Y |
| Robinson et al. (Rob87) [11] | $S(x,y) = \frac{y-x}{y+x} \cdot 200$ | 0 | -200,200 | Y |
| Vagenas and Hoshizaki (Vag92) [12] | $S(x,y) = \frac{y-x}{\max\,(x,y)} \cdot 100$ | 0 | -100,100 | Y |
| Plotnik et al. (Plo05) [13] | $S(x,y) = \left|\ln\left(\frac{y}{x}\right)\right| \cdot 100$ | 0 | $0,\infty$ | N* |
| Zifchock et al. (Zif08) [14] | $S(x,y) = \left(\frac{45° \,-\text{atand}\,\left(\frac{x}{y}\right)}{90°}\right) \cdot 100$ | 0 | -50,50 | Y |
| Rochester et al. (Roc14) [16] | $S(x,y) = \|y - x\|$ | 0 | $\infty,\infty$ | N* |
| Queen et al. (Que20) [17] | $S(x,y) = \frac{y-x}{\max\,(0,x,y) - \min\,(0,x,y)}$ | 0 | -1,1 | Y |
| Alves et al. (Alv20) [18] | $S(x,y) = \frac{y-x}{\sqrt{2\left(x^2+y^2\right)}}$ | 0 | -1,1 | Y |
| Alves et al. (Alv20b) [18] | $S(x,y,\sigma) = \frac{y-x}{\sqrt{2\left(x^2+y^2\right)}} \cdot \left(1 - \frac{\sqrt{2}\sigma}{\sqrt{2\sigma^2+x^2+y^2}}\right)$ | 0 | -1,1 | Y |

*Removing the absolute value enables directional output.

[†]These limits are accurate for inputs of the same sign, however, many of the discrete symmetry metrics can produce larger values for inputs of differing signs (not including Plotnik et al. [13], where the logarithm requires identically signed inputs).

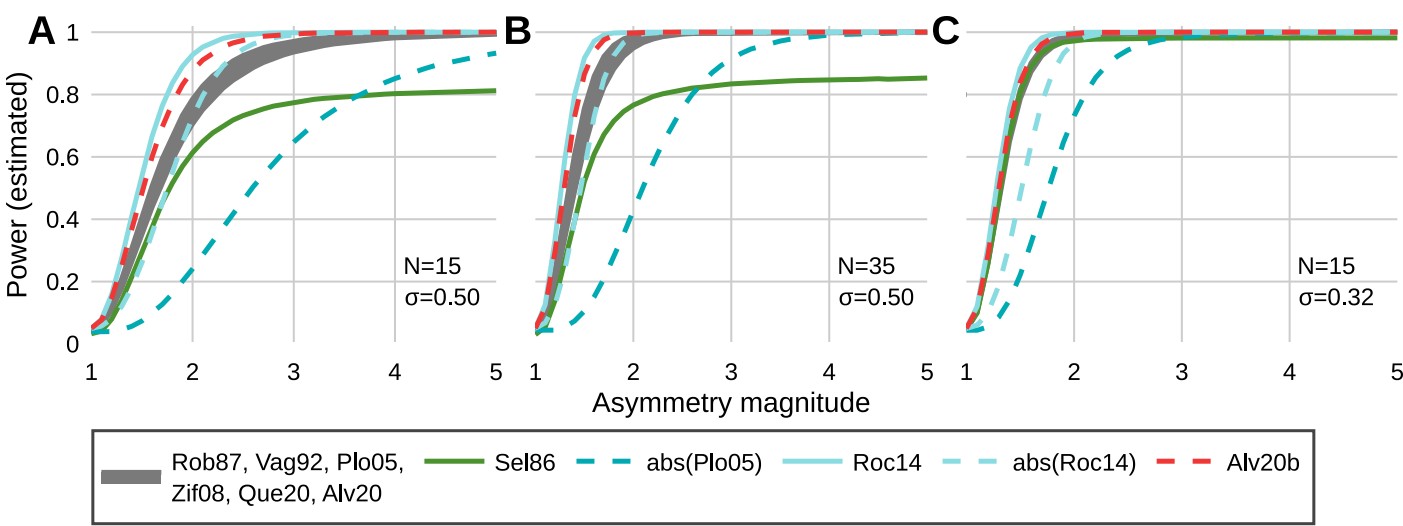

**Fig 1. Estimated power vs. asymmetry magnitude.** Estimated power for (A) a sample size of 15 and a variability of 0.50. (B) a sample size of 35 and a variability of 0.50. (C) a sample size of 15 and a variability of 0.32.

value for every metric. Symmetry metrics are abbreviated as the first three letters of the first author followed by the last two digits of the publication year, and metrics with an absolute value are distinguished by an absolute value function annotation, e.g. "abs(Plo05)".

## Power simulation

As sample size increases and group/condition variability decreases, all metrics reach sufficient power at a lower asymmetry magnitude (Fig 1A vs 1B and 1A vs 1C). When $R = 1$ (a true null effect), average power was approximately equal to the critical alpha, $0.049 \cong \alpha$. Six metrics (Rob87, Vag92, Plo05, Zif08, Que20, and Alv20) display practically equivalent sensitivity for all variabilities and sample sizes (Fig 1). The ratio (Sel86) ceases to asymptotically approach 100% power for variability greater than 0.25, regardless of sample size; at the largest sample size (100) and variability (1.0), sufficient power was not reached for the largest asymmetry magnitude (power = 0.79). In contrast, the abs(Plo05) metric approached 100% power for the entire range of variabilities, but increased in power much slower than the ratio. The slow increase in power of the Sel86 and abs(Plo05) metrics compared to all other metrics was practically significant for all sample sizes, and increased with variability (Figs 1 and 2). The abs(Plo05) and abs(Roc14) metrics both show large decreases in power compared to their respective non-absolute valued versions and increase in power with asymmetry magnitude much slower in comparison (Fig 1). The Roc14 and Alv20b metrics reached sufficient power quicker than other metrics for all variabilities and sample sizes, however meaningful differences between these and other metrics only exist for variability greater than 0.25. The Alv20b metric (with the weighting factor) increases in power faster than the unweighted (Alv20) version, however, this difference is only practically relevant for variabilities greater than 0.5.

## Experimental results

Gait trials from fifteen healthy, young adults (7 female, 23.4 ± 2.8 years (mean ± s.d.); 72.3 ± 13.5 kg; 170.2 ± 8.1 cm) were analyzed [23]. Table 2 reports the average and standard deviation of the swing time, calculated from 140 strides per trial, and the equivalent magnitude

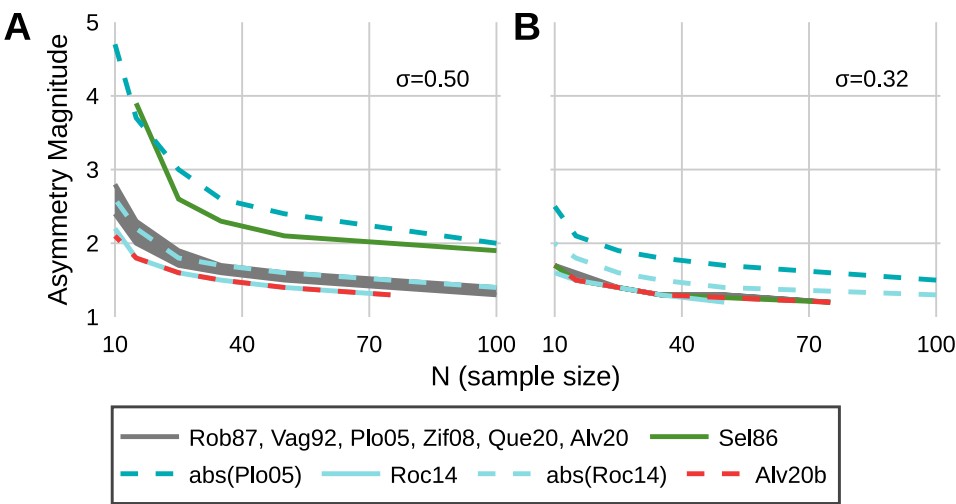

**Fig 2. Minimum sample size to achieve sufficient power (80%).** Shown at a variability of (A) 0.50 and (B) 0.32. The maximum and minimum sample tested were 100 and 10, respectively. Lower asymmetry magnitude and smaller N (i.e. closer to the bottom left corner) represents higher sensitivity.

asymmetry and variability in the format used in the power simulations. Left and right swing times for both tied and split-belt conditions passed the Anderson-Darling test of normality ($p > 0.46$). Both tied and split-belt conditions passed Bartlett's test of equality of variances between the left and right sides ($p > 0.44$). Results for swing time asymmetry evaluated with all metrics are reported in Table 3. All metrics produced significant differences ($p < .001$).

## Discussion

The results of the power simulation follow several basic principles of statistical power: power for a true null effect is equivalent to the critical alpha, power increases with effect size (e.g. increases in asymmetry magnitude and decreases in variability) and sample size [21]. Our results confirm that symmetry metrics assessed in this study exhibit different power for the same underlying effects of asymmetry, particularly for small effects. Two metrics (Roc14 and Alv20b) were highly sensitive and robust to increased variability, two metrics (Sel86 and abs (Plo05)) show poor sensitivity to small effects and high variability. Finally, six metrics (Rob87, Vag92, Plo05, Zif08, Que20, Alv20) exhibited similar sensitivity, at slightly less than the non-absolute value Roc14 and Alv20b metrics. Additionally, the swing time asymmetry results validated assumptions made for the power simulation (independent and normally distributed inputs with equal variances) and were situated within the range of asymmetry magnitudes, variabilities, and sample sizes tested in the power simulation.

The use of the split-belt treadmill to mechanically induce asymmetric changes in swing time prevents the occurrence of false positive test statistics and allows greater confidence in assessing sensitivity based on differences in findings of significance among the metrics. Despite

**Table 2. Swing time.**

| Condition | Left (%) | Right (%) | Asymmetry magnitude (R) | Variability (σ) |
|---|---|---|---|---|
| Tied-belt | 38.6 ± 0.67 | 38.9 ± 0.67 | 1.01 | 0.017 |
| Split-belt | 40.5 ± 1.3 | 36.2 ± 1.1 | 1.12 | 0.033 |

**Table 3. Swing time asymmetry.**

| Metric | Mean difference* | 95% CI | | t(15) | $d_{unbiased}$ |
|---|---|---|---|---|---|
| | | Lower | Upper | | |
| Seliktar and Mizrahi [10] | -0.114 | -0.131 | -0.0963 | -14.2 | 4.03 |
| Robinson et al. [11] | -12.0 | -14.0 | -10.0 | -13.0 | 3.80 |
| Vagenas and Hoshizaki [12] | -11.3 | -13.1 | -9.61 | -14.1 | 4.03 |
| Plotnik et al. [13] | -12.0 | -14.0 | -10.0 | -12.9 | 3.79 |
| Plotnik et al. [13] (abs) | 10.1 | 7.47 | 12.7 | 8.29 | 3.25 |
| Zifchock et al. [14] | -3.82 | -4.44 | -3.19 | -13.0 | 3.81 |
| Rochester et al. [16] | -4.61 | -5.39 | -3.82 | -12.6 | 3.71 |
| Rochester et al. [16] (abs) | 3.86 | 2.83 | 4.88 | 8.09 | 3.17 |
| Queen et al. [17] | -0.113 | -0.131 | -0.0961 | -14.1 | 4.03 |
| Alves et al. [18] | -0.0599 | -0.0697 | -0.0501 | -13.1 | 3.82 |
| Alves et al. [18] (weighted) | -0.0575 | -0.067 | -0.0481 | -13.0 | 3.82 |

*Mean difference refers to the average of the pair-wise group difference between tied- and split-belt walking conditions.

this, several aspects of the swing time results demonstrate the need for power simulations to assess metric sensitivity. First, the large effect sizes demonstrate that every metric was highly powered to detect the effect of asymmetry in swing time, and the unanimous findings of significance for this large effect of asymmetry does not preclude disagreement at smaller effect sizes (i.e. any differences in metric sensitivity are ambiguous in this data). This ambiguity in metric sensitivity due to large effect sizes is similarly apparent when effect size (Cohen's d) is manually calculated from several gait variables in the results of Patterson et al. [19, Table 1]. Second, all things being equal (sample size, significance criterion), power is a direct function of effect size [21]. However, the inherent uncertainty of observed effect sizes is emphasized by the mismatch between the order of effect sizes in Table 3 and the general results of the power simulation: The abs(Plo05) and Roc14 metrics exhibit the two smallest effect sizes for swing time, despite the power simulation showing large differences in power between these metrics for small asymmetry magnitudes with non-negligible variability. Similarly, the ratio (Sel86) produced the largest effect size, which could falsely support an interpretation of high sensitivity, but a key behavior of the ratio (lack of robustness to high variability compared to other metrics, see Fig 1) is not apparent in the swing time results, due to the low group variability in swing time. The poor sensitivity of the Sel86 metric in the presence of non-negligible variability is due to the definition of symmetry as a ratio. The distribution of a ratio of two normally distributed random variables is a Cauchy distribution—which has an analytically undefined variance. In practice, this can lead to a large variance that obscures mean differences that might otherwise be significant.

In contrast, the power simulation suggests that the Rochester et al. [16] metric without the absolute value, a simple difference, is the most sensitive, with a caveat that it is the only metric which is not normalized by a reference value. The weighted Alves et al. [18] metric is a normalized alternative that has practically equivalent sensitivity. All other symmetry metrics evaluated here produce relative bilateral differences, such that the asymmetry of $S(x, y) \neq S(x + c, y + c)$; this is also called "scale invariance" by Alves et al. [18] who asserted that symmetry metrics should display this behavior. The lack of scale invariance by the Roc14 metric may be mitigated in some cases via the addition of a covariate to a statistical test. Alternately, all the other metrics assessed here [11, 12, 14, 17, 18] are scale invariant, practically equivalent in sensitivity

to effects of asymmetry, and only slightly less sensitive than the metric proposed by Rochester et al. [16].

A strength of this study is that the parameters (asymmetry magnitude and variability) of the power simulation are the major factors in effect sizes, and therefore the power simulation essentially simulated different effect sizes. This allows the results of the power simulation to generalize to more statistical tests than the paired t-test used here. However, real data may not exhibit all the characteristics (asymmetry magnitudes and variability) and assumptions (independent and normally distributed inputs with equal variances, asymmetric data compared to symmetric baseline). Our simulation code and results are available in Zenodo and can be further explored or expanded to test characteristics or assumptions not made here [27].

## Recommendations

The results of this study show that multiple symmetry metrics demonstrate sufficient sensitivity for a broad range of data. However, several practices may reduce the isolation of results based on the numeric differences in the results of various symmetry metrics. First, reporting of bilateral data in addition to symmetry metric results enables the direct calculation of asymmetry with alternate symmetry metrics and aids in comparisons to other studies and/or populations. Second, in the context of a population analyzed using affected/unaffected sides instead of left/right, reporting the bilateral data for both affected/unaffected and left/right, along with the correlation between affected side and limb dominance, communicates valuable information that cannot be otherwise inferred. Third, in agreement with the conclusions of Patterson et al. [19], a ratio is a more intuitive format for reporting results than other metrics. To improve the communication of results while maintaining a higher power for statistical testing, a metric with good sensitivity could be used for the statistical analysis, and then the results of the analysis—means and confidence intervals—could be transformed to ratios for reporting; such an approach would combine the strengths of a more sensitive metric and the intuitiveness of a ratio. Confidence intervals would need to be used because the lower and upper bounds can be exactly transformed between metrics, while differences in how each metric affects variability prevents the conversion of standard deviations between metrics.

## Conclusion

In this study, we compared commonly used discrete symmetry metrics using power simulations and real data to demonstrate that metrics exhibit different sensitivities to the same underlying effects of asymmetry. Two metrics, published by Rochester et al. [16] (when used without an absolute value) and Alves et al. [18], display excellent sensitivity to a broad range of data characteristics. However, some metrics display very poor sensitivity when data is highly variable, therefore we suggest that future studies consider metric sensitivity to reduce the possibility of underpowered research.

## Acknowledgments

The authors would like to thank Tarique Siragy and Mary-Elise MacDonald for proof-reading drafts of this manuscript.

## Author Contributions

**Conceptualization:** Allen Hill.

**Data curation:** Allen Hill.

**Formal analysis:** Allen Hill.

**Funding acquisition:** Julie Nantel.

**Investigation:** Allen Hill.

**Methodology:** Allen Hill.

**Project administration:** Julie Nantel.

**Resources:** Julie Nantel.

**Software:** Allen Hill.

**Supervision:** Julie Nantel.

**Validation:** Allen Hill.

**Visualization:** Allen Hill.

**Writing – original draft:** Allen Hill.

**Writing – review & editing:** Allen Hill, Julie Nantel.

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
