## [Decision Letter · Decision Letter 0]

16 Feb 2022

PONE-D-21-30790Sensitivity of discrete symmetry metrics: implications for metric choicePLOS ONE

Dear Dr. Nantel,

Thank you for submitting your manuscript to PLOS ONE. After careful consideration, we feel that it has merit but does not fully meet PLOS ONE’s publication criteria as it currently stands. Therefore, we invite you to submit a revised version of the manuscript that addresses the points raised during the review process.

We look forward to receiving your revised manuscript.

Kind regards,

Tuhin Virmani

Academic Editor

PLOS ONE

Journal Requirements:

Reviewers' comments:

Reviewer's Responses to Questions

**Comments to the Author**

1. Is the manuscript technically sound, and do the data support the conclusions?

Reviewer #1: Yes

Reviewer #2: Yes

2. Has the statistical analysis been performed appropriately and rigorously? 

Reviewer #1: Yes

Reviewer #2: I Don't Know

3. Have the authors made all data underlying the findings in their manuscript fully available?

Reviewer #1: Yes

Reviewer #2: Yes

4. Is the manuscript presented in an intelligible fashion and written in standard English?

Reviewer #1: Yes

Reviewer #2: Yes

5. Review Comments to the Author

Reviewer #1: Main feedback to the author

The authors have investigated the sensitivity of different discrete symmetry metrics available in the literature. The topic is of interest to help researchers, clinicians, and other professionals of interest to investigate symmetry as, for example, a recovery indicator based on a sound and robust methodology. In this study, the authors have first considered a theoretical approach to investigate the sensitivity of the included discrete symmetry metrics. In a second step, gait data was considered to experimentally verify the robustness of the discrete symmetry metrics investigated. This is a very elegant and valid approach. However, considering the broad range of signals available in both healthy and pathological populations exhibiting asymmetric characteristics, restricting the analyses only to bilateral signals with the same sign, e.g., positive, potentially is not the most representative approach. The current analyses exclude relevant signals, e.g., kinetic data, which could contribute to understanding better the asymmetry in multiple populations. I strongly suggest the authors reconsider this and the aspects described below to improve the manuscript quality.

The following comments need to be addressed:

Major issues

Line 114: I would appreciate it if the authors could comment on the definition of R = [1,5]. This approach only includes variables with the same sign (x and y > 0, for example), which is not fully representative of kinetic data, for example. I would agree with this definition if the authors would only be interested in spatiotemporal aspects (or any other signal only with positive OR negative signs). If this is the case, the type of signal of interest must be referred to on line 96, i.e., x and y are considered to always have the same sign. However, if the authors choose this strategy the usability of this manuscript will be limited and it should be included in the limitations. I would suggest including another range that crosses zero and experimental data that reflects it (e.g., ground reaction forces in the anteroposterior direction, which may be used in stroke populations – the authors refer to this population in line 38).

General comment for the results section: it is difficult to follow the figures and the results section with the multiple references and labels used in both the text and the figures’ captions. Please be consistent: either use the abbreviations you have in the figures or use the references in the figures. I would strongly advise using abbreviations as a simple and efficient solution. This should ideally be introduced in the Methods section (line 104). Please also refer to the figures’ panels (e.g., Figure 1 A) when describing the results within the text.

Line 273: I recommend caution with this statement since the authors did not investigate other types of signals. It is currently unknown what would happen in scenarios that lead to “artificial inflation”. Furthermore, considering the results described in lines 163 – 166, this recommendation could mislead future results comparisons.

Minor issues

Line 45: While I understand what the authors refer to, I would suggest rethinking the use of "discrete symmetry metric". Potentially, a reference to “method” instead of “metric” could improve manuscript readability. Why did the authors use the abbreviation DSM only in the introduction? I would suggest using DSM across the whole text or using only the long-form.

Line 104: Please refer to the discrete symmetry metrics, potentially with abbreviations as done in the figures’ captions. This will help the manuscript's readability.

Lines 149 – 151: Please specify in which discrete symmetry metrics the order of x and y was adjusted.

Lines 158 – 160: I assume the authors are comparing Figure 1 A to Figure 1 B and Figure 1 A to Figure 1 C. However, it would be helpful if the authors could guide better the reader through this result. This is a relevant figure to understand the sensitivity of the different discrete symmetry metrics and the discussion/findings.

Line 163: please be consistent with the range you report. A range of [0,1] (in figures for example) and [0,100] may represent the same but it should be reported consistently.

Line 164: why did the authors use a value on line 164 (0.25) and another in the figures (0.32)?

Lines 164 – 166: where can the reader see this result? It would be helpful if the authors could explain this better.

Lines 166 – 168: I assume that “the absolute value Plotnik et al. [13]” refers to abs(Plo05). From what I can see, abs(Plo05) did not reach 100% power for all variabilities (Figure 1 A).

Lines 168 – 170: How can we see this effect (slow increase in power) in Figure 2 if Figure 2 refers to “Minimum sample size to achieve sufficient power (80%)”? Is this not supposed to be read together with Figure 2?

Lines 168 – 170 and 173 – 175: The authors report in the results section that some discrete symmetry metrics increase in power faster than others. Considering this, I would expect the authors to have discussed this in this section. What could be consequences (and even recommendations) following such a result?

Lines 175 – 176: where can this be observed in Figure 1? The three panels refer to variability greater than 0.25.

Line 192 – 193: throughout the manuscript, the reader can read “swing time proportion”, “swing time percent” (line 135), and only “swing time”. Keeping it consistent would help the manuscript’s readability.

Lines 197 – 198: the authors did not report swing time asymmetry evaluated with all metrics in Table 3. The mean difference was reported. Please keep this consistent and explain what the mean difference refers to.

Line 202: Swing time (%) proportion asymmetry or swing time (s) asymmetry? Please be consistent throughout the manuscript.

Figures: the figures are beautiful. Consistent, good choice of colors, and appropriate size. I like very much the strategy to aggregate the discrete symmetry metrics with very similar results – it improves the figure a lot! I only have very minor suggestions:

Figure 1: It would be very helpful if the authors could include in each panel the sample size and the variability used.

Figure 2: It would also be very helpful if the authors could include in each panel the variability used.

Table 3: Expanding the captions would be of great help for the reader. Your paper could be better understood, in my perspective. For example, briefly describe what mean difference refers to.

Lines 209 – 215: this is results repetition (lines 160 – 163, for example) – all valid and important results. Considering the large number of investigations that are based on some of the discrete symmetry metrics tested in this study that had poor sensitivity (e.g., the ratio [10]) or similar sensitivity (e.g., Robinson et al. [11], Zifchock et al. [14]), the authors could expand this.

Lines 243 – 244: from what I can observe in Figure 2, Rochester et al. [16] is not the most sensitive. Rochester et al. [16] and the weighted Alves et al. [18] have the same sensitivity. I could also not find this result before. Please reconsider either here or in the results section to keep consistency.

Line 262: What do the authors mean by “continued use of multiple symmetry metrics”? To use multiple discrete symmetry metrics without considering their sensitivity? This would be opposite to lines 272 – 273.

Reviewer #2: The authors verified the sensibility of metrics employed by the literature to calculate gait (a)symmetry. The study has the potential in supporting further studies in selecting the most appropriate metrics to examine gait asymmetry. Although, in my opinion, the study might be relevant, I think some segments of the manuscript should be strengthened. For instance, even though it is not the intention, the authors should explain in more detail the clinical/functional relevance of studying gait asymmetry. Mainly, what I am struggling with is the end segment of the study, since, considering the title (“Sensitivity of discrete symmetry metrics: implications for metric choice”), I was expecting a recommendation in terms of the most appropriate metric to select to calculate gait symmetry. If it is not possible because metrics may have pros and cons, I would recommend a list mentioning these pros and cons aspects to guide future research in the field.

Specific comments

- Introduction - Across paragraph 1

The authors introduced the topic and briefly discussed clinical aspects of gait asymmetry. I believe it is possible to strengthen by including the clinical and functional relevance of examining gait asymmetry. Perhaps these questions might help: In gait study, what is the implication of gait asymmetry? If gait asymmetry is implicit in humans, why is important to reduce gait asymmetry in the neurological population? Also, briefly, what are the physiological, neural, and/or biomechanics underlying mechanism related to gait asymmetry that makes this topic relevant to study?

Introduction paragraph across lines 48-59

- I am confused with the information in this paragraph;

1. the authors referred to 5 discrete methods. However, if I understood it correctly, there were 8 described metrics (DSMs) (Refs. 10, 11, 12, 13, 14, 15 and 16, 17 and 18) So, was it 5 or 8, or Did I miss something?

2. As the information is described in this paragraph, I could not understand whether there are differences and overlaps between the matrics. I recommend the authors review the paragraph clarifying in more detail what made the metrics different. I

3. The authors should call table 1 here and maybe include more information in the table to discriminate or link the information of the metrics

Methods, lines 106 – 108 “Two metrics [13,16] are defined with absolute values and solely assess asymmetry magnitude; to enable consistent comparisons between metrics, these two metrics were assessed with and without the absolute value applied.”

- I believe being important to explain better this segment, meaning what did the authors mean to say using absolute value and magnitude? how do the studies differentiate that?

Experiment results

- If it is possible, as the authors addressed the relevance of power, I strongly recommend the authors to include a column in tables indicating the current power for the comparisons.

Recommendation sections – Discussion

- In my opinion, as a recommendation, I suggest including a bulleted list indicating the main points found by the authors. This could help guide further study in gait asymmetry.

Discussion

Somehow, to me, it would be important to readers if the authors could include some discussion about the clinical implication of selecting one or another metric of gait asymmetry

Discussion

- I recommend including a limitation subsection. I noticed that the authors described some limitations across the discussion, but I strongly recommend summarizing the limitation in a specific section.

Conclusion

- In my opinion, the conclusion needs to be improved in terms of indicating the most appropriate metric(s), as it was implicitly mentioned in the title.

6. PLOS authors have the option to publish the peer review history of their article (what does this mean?). If published, this will include your full peer review and any attached files.

Reviewer #1: No

Reviewer #2: No

---

## [Author Response · Author response to Decision Letter 0]

8 Mar 2022

A response to reviewers file has been uploaded with the updated manuscript

---

## [Decision Letter · Decision Letter 1]

26 Apr 2022

PONE-D-21-30790R1Sensitivity of discrete symmetry metrics: implications for metric choicePLOS ONE

Dear Dr. Nantel,

Thank you for submitting your manuscript to PLOS ONE. After careful consideration, we feel that it has merit but does not fully meet PLOS ONE’s publication criteria as it currently stands. Therefore, we invite you to submit a revised version of the manuscript that addresses the points raised during the review process.

We look forward to receiving your revised manuscript.

Kind regards,

Tuhin Virmani

Academic Editor

PLOS ONE

Journal Requirements:

Reviewers' comments:

Reviewer's Responses to Questions

**Comments to the Author**

1. If the authors have adequately addressed your comments raised in a previous round of review and you feel that this manuscript is now acceptable for publication, you may indicate that here to bypass the “Comments to the Author” section, enter your conflict of interest statement in the “Confidential to Editor” section, and submit your "Accept" recommendation.

Reviewer #1: All comments have been addressed

Reviewer #2: All comments have been addressed

2. Is the manuscript technically sound, and do the data support the conclusions?

Reviewer #1: Yes

Reviewer #2: Yes

3. Has the statistical analysis been performed appropriately and rigorously? 

Reviewer #1: Yes

Reviewer #2: Yes

4. Have the authors made all data underlying the findings in their manuscript fully available?

Reviewer #1: Yes

Reviewer #2: Yes

5. Is the manuscript presented in an intelligible fashion and written in standard English?

Reviewer #1: Yes

Reviewer #2: Yes

6. Review Comments to the Author

Reviewer #1: Thank you for your effort to address all my comments - they were adequately addressed and the quality of the manuscript is improved. I have no further comments.

Reviewer #2: The authors answered my inquiries. Although I agree that (as I highlighted in my previous comments) the clinical relevance of the metrics is not part of the aim, I believe clinicians and researchers-related will not only make the decision about metrics only based on sensibility but also on what is the clinical relevance of the metrics for their population of interest. This is why I do believe it was important to strengthen the clinical relevance of the metrics in the first place since it would increase further the potential of the manuscript in supporting studies with selecting asymmetry metrics. However, as I mentioned, I agree it is not part of the aim, and I am okay with the authors’ response and decision to not include it.

My only specific comment is with the conclusion (line 280-281) in which I think the authors should add a note reinforcing that the metric of Rochester et al., [16] displayed excellent sensitivity when the absolute value was not considered. Then, the information here would be consistent with the argument in lines (240-241). If the authors agree, a suggestion can be:

“Two metrics, published by Rochester et al. [16] (without considering absolute value) and Alves et al. [18], display excellent sensitivity to a broad range of data characteristics.

7. PLOS authors have the option to publish the peer review history of their article (what does this mean?). If published, this will include your full peer review and any attached files.

Reviewer #1: No

Reviewer #2: No

---

## [Author Response · Author response to Decision Letter 1]

29 Apr 2022

We thank reviewer 2 for their continued dedication to improving the content and communication of our study. We have accepted (with a minor rephrasing) the suggested modification to the sentence (line 280-281) in the conclusion. The sentence now reads as “Two metrics, published by Rochester et al. [16] (when used without an absolute value) and Alves et al. [18], display excellent sensitivity to a broad range of data characteristics.”

---

## [Editor Report · Decision Letter 2]

3 May 2022

Sensitivity of discrete symmetry metrics: implications for metric choice

PONE-D-21-30790R2

Dear Dr. Nantel,

We’re pleased to inform you that your manuscript has been judged scientifically suitable for publication and will be formally accepted for publication once it meets all outstanding technical requirements.

Kind regards,

Tuhin Virmani, MD, PhD

Academic Editor

PLOS ONE

---

## [Editor Report · Acceptance letter]

11 May 2022

PONE-D-21-30790R2 

Sensitivity of discrete symmetry metrics: implications for metric choice 

Dear Dr. Nantel:

I'm pleased to inform you that your manuscript has been deemed suitable for publication in PLOS ONE. Congratulations! Your manuscript is now with our production department. 

Kind regards, 

on behalf of

Dr. Tuhin Virmani 

Academic Editor

PLOS ONE